# Distributed Acoustic Sensing Based on Microtremor Survey Method for Near-Surface Active Faults Exploration: A Case Study in Datong Basin, China

**DOI:** 10.3390/ijerph20042915

**Published:** 2023-02-07

**Authors:** Ao Song, Junjie Ren, Aichun Liu, Guangwei Zhang, Xiaoqiong Lei, Hao Zhang

**Affiliations:** 1National Institute of Natural Hazards, Ministry of Emergency Management of China (MEMC), Beijing 100085, China; 2Key Laboratory of Compound and Chained Natural Hazards Dynamics, MEMC, Beijing 100085, China; 3Institute of Geomechanics, Chinese Academy of Geological Sciences, Beijing 100081, China

**Keywords:** microtremor survey, near-surface active fault exploration, distributed acoustic sensing, multichannel analysis surface wave

## Abstract

Active fault detection has an important significance for seismic disaster prevention and mitigation in urban areas. The high-density station arrays have the potential to provide a microtremor survey solution for shallow seismic investigations. However, the resolution limitation of the nodal seismometer and small-scale lateral velocity being inhomogeneous hinder their application in near-surface active fault exploration. Distributed acoustic sensing (DAS) has been developed rapidly in the past few years; it takes an optical fiber as the sensing medium and signal transmission medium, which can continuously detect vibration over long distances with high spatial resolution and low cost. This paper tried to address the issue of near-surface active fault exploration by using DAS. We selected a normal fault in the southern Datong basin, a graben basin in the Shanxi rift system in north China, to carry out the research. Microtremor surveys across the possible range of the active fault were conducted using DAS and nodal seismometers, so as to obtain a shallow shear wave velocity model. Meanwhile, we applied a Brillouin optical time domain reflectometer (BOTDR) and distributed temperature sensing (DTS) to monitor the real-time fluctuation of ground temperature and strain. Our results show that the resolution of the deep structures of the fault via the microtremor survey based on DAS is lower than that via the seismic reflection; whereas, their fault location is consistent, and the near-surface structure of the fault can be traced in the DAS results. In addition, both the BOTDR and DTS results indicate an apparent consistent change in ground temperature and strain across the fault determined by the DAS result, and the combination of surface monitoring and underground exploration will help to accurately avoid active faults and seismic potential assessment in urban areas.

## 1. Introduction

Large earthquakes occurring in urban areas often bring about catastrophic events and cause enormous casualties and huge economic losses in many seismically active regions [1]. For example, the 1976 Ms 7.8 Tangshan earthquake in north China destroyed the entire city and killed over 240,000 [2]. The 1995 Mw 7.3 Kobe earthquake in the southern part of Hyōgo Prefecture, Japan, broke the bridges and roads and resulted in 6434 dead [3]. In addition, in 2010, a catastrophic magnitude Mw 7.0 earthquake struck Port-au-Prince, Haiti’s capital, and claimed more than 250,000 lives [4]. The main reason for these catastrophic events is that the causative faults of these earthquakes are buried directly below the urban areas. Therefore, searching for beneath active faults with the potential of large earthquakes beneath a city becomes a crucial step for urban seismic hazard assessment and disaster prevention and migration [5]. The active source seismic reflection technique has been proven to be the best technique in the exploration of active faults for its high resolution and deep penetration of fault structures [6,7]. However, the high cost and strong dependence of this technique make it hard to widely use in urban complex environments. In recent years, the great advancements in nodal seismometers have boosted the growth of microtremor surveys [8]. Although their accuracy and resolution are lower than those of active seismic surveys, they have the advantage of microtremors and of being a low-cost technique, and have been verified to successfully image the detailed crystal structures [9] and subsurface exploration [10,11]. The resolution of microtremor surveys depends on the frequency content of microtremors and the spatial sampling interval, which drives microtremor surveys’ data acquisition toward a “larger and dense” geometry. However, nodal seismometers enable high-density spatial sampling at the cost of eliminating small arrays, which means that it is difficult to apply the nodal seismometers to large-scale urban planning and active fault avoidance.

DAS is a newly developed optical fiber sensing technology that uses the coherent Rayleigh backscattering of a low-noise laser in a common single-mode sensing fiber to continuously detect the vibration variation of an external physical field over a long distance [12,13]. To date, the application in downhole, marine, and surface seismic data acquisition has shown that this technology is a compelling alternative to modern point-sensor acquisitions with low operating costs, a broad frequency response, small spatial sampling intervals and a high number of intrinsically synchronized channels [14,15,16], and it has obtained good results in the seismology community such as in earthquake detection [17,18,19,20], hydrocarbon and geothermal exploration [21,22,23], urban monitoring [24,25,26], volcano and glacier monitoring [27,28,29], subsurface imaging [30,31,32,33,34,35,36,37] and other seismic activities [38,39]. However, the applicability of this technique in the explorations of active faults remain unknown.

In this paper, we conduct a near-surface active fault detection experiment based on microtremor surveys using DAS in the southern suburb of Datong basin, north China. According to our seismic reflection profiles and existing geological surveys, we chose a test site and deployed optic cables to test the application of the distributed optical fiber sensing in the exploration of active faults. To verify the authenticity and validity of DAS data, we also deployed short-period three-component seismometer lines. First, we describe the major tectonic features of the test site based on previous studies. Second, we compared the microtremor data recorded by DAS arrays and three-component nodal seismometers and applied multichannel analysis of surface waves (MASW) [40] to identify the location of the near-surface active fault. Finally, we validate the inversion results with reference to the active source seismic reflections, and discuss the relationship between the ground temperature and strain data recorded by DTS and BOTDA and the location of near-surface active faults. The results show that the DAS microtremor surveys together with the ground temperature and strain recorded by DTS and BOTDR can be taken as a comprehensive and inexpensive supplementary technique to image near-surface activated faults in urban areas. This study extends the distributed optical fiber sensing, showing the practical benefits of low cost and high resolution to explore the active faults in urban areas.

## 2. Geological Setting

The Datong basin is a graben in the northern part of the Shanxi rift system that is characterized by a series of basin boundary normal faults and several large earthquakes with a magnitude larger than 7 in history [41]. No large earthquakes hit the Datong basin; although, there is a historic record of over 2000 years long. Figure 1a shows the geological map of Datong basin, which is located in the northern Datong basin. The Kouquan fault in the west and the Liulengshan fault in the east represent the west and east boundary of the Datong basin. Geological observations demonstrated that these boundary faults are primarily normal and have experienced several paleo-earthquakes in the late Quaternary [42]. Our seismic reflection profiles in the exploration of active faults for Datong City indicate that there are numerous secondary active faults possibly buried by the Quaternary sand and soil layers of the basin (as shown by the black solid line in Figure 1a). 

Fault F4 is the boundary normal fault of the sub-basin and dips NW; the seismic profile shown in Figure 1b is the SE–NW reflection seismic control line XL previously arranged by our research group for seismic safety assessment of the Datong basin in June 2011, which crosses the F4 fault. We can see that the lateral variation in the profile XL is large, and the stratigraphic interface is generally characterized by ladder-shaped sedimentation with shallow northwestern and deep southeastern layers. The two-way travel time of effective reflected information is approximately 1.3 s. The seismic reflection results indicate that the fault might reach the ground surface, implying that fault F4 might have the potential of a surface-rupturing earthquake, but accurately determining the near-surface location of the fault is still difficult. In addition, the area between Datong City and the Liuleng Mountain piedmont fault is a future urban development area. However, the existing data show that earthquakes in the Datong basin are mainly distributed near the Liuleng Mountain piedmont fault, and an accurate detection of active faults could be helpful with respect to earthquake hazard assessment and safety design in urban planning. Again, this experiment area has a flat terrain, convenient transportation, and multiple types of surface environments, such as farmland, towns and shelter forests, which is conducive to study the effects of different external factors on the distributed optical fiber. 

Therefore, we selected the area where fault FP4-3 is located as the test site (the red dashed box in Figure 1b). The fault is a normal fault extending to the near-surface with an apparent NW dip, and the upper breakpoint is located near station 7000 m. Reflected event dislocation occurred on both sides of the fault. On the northwest side of the fault, the events obviously rise, the number increases, and there is an obvious bedrock interface on the southeast; these features are conducive to the identification of faults.

## 3. Field Observations across the Fault

Our survey line is close to the road, which is conducive to microtremor surveys [43]. The optical fiber used in this experiment is a conventional single component communication optical cable. DAS, BOTDR and DTS are used to record microtremor, ground strain and temperature (L1 in Figure 2a), respectively, and 128 units of 0.5 Hz three-component nodal seismometers were laid out along the optical fiber as a reference group (L2 in Figure 2a). Limited by the number of nodal seismometers, the spacing is small (5 m) in the areas where the active fault may pass through and large (10 m) in other areas. To keep the optical fiber and seismometers consistent in terms of spatial points, we set up 250 optical fiber sensors with an interval of 4 m. The DAS interrogator used in this paper was developed by Liu Aichun, a senior engineer who is the third author of this paper (Figure 2b). As shown in Figure 2c, we dug a horizontal trench with approximately 10 cm depth to bury nodal seismometers and an optical fiber, and a tamping machine shown in Figure 2d was activated on the road 1 m away from the optical cable to test the vibration signal received by seismometers and DAS. The seismic data acquisition started simultaneously and continued for 5.1 h with sampling rates of 8000 Hz and 400 Hz, respectively. The specific geometric parameters of the microtremor surveys are shown in Table 1.

## 4. Data Analysis and Processing

Figure 3 presents the raw microtremor data acquired at the test site by DAS and the three-component nodal seismometers array; the signals excited by the tamping machine are clearly observed (Figure 3a). The energy of the DAS signal, with frequency of about 65 Hz, is much larger than the signal recorded by the nodal seismometers; these strong energy signals are the fading noise, which are caused by the destructive interference of Rayleigh backscattering in the sensing fiber and do not contain underground information. We need to filter these interferences in the subsequent processing. For the main signal frequency band (4–150 Hz), we find a good match of the microtremor power spectral density between the three-component nodal seismometers and DAS data, except at frequencies below ~4 Hz where the noisy power spectral density diverges (Figure 3b). A contributing factor to this divergence is probably that the signal-to-noise ratio (SNR) of DAS data deteriorates quickly at low frequencies.

Before calculating the multi-offset cross-correlation (CC), we used the three-component microtremor data recorded by the nodal seismometers to calculate the H/V spectral ratio curves. This method is generally used to provide prior information for subsequent microtremor survey data processing [44], and we mainly determine the length of DAS array used in the CC calculation based on the H/V spectral ratio curves. Figure 4 presents the calculation result, in which the top horizontal axis is the distance in meters, and the vertical axis is the dominant frequency ratio of the horizontal and vertical components. It shows that the H/V spectral ratio curve has only one peak frequency of 1 Hz in the north and two peak frequencies of 1 Hz and 2 Hz in the south. The peak frequency in the H/V spectral ratio curve is very close to the dominant frequency of the sedimentary layer [45], and the dominant frequency and depth of the bedrock surface can be calculated by the empirical formula shown in Formula (1) [46]:
(1)H=Vs/4f

f and H represent the dominant frequency and thickness of the soil layer, respectively, Vs is the S-wave velocity. Therefore, there may be a buried interface that becomes shallower toward the southern region. According to the obvious 1 HZ peak frequency in the H/V spectral ratio curve, we speculate that its depth is about 150 m. According to empirical Formula 1, we choose the initial arrangement length of 300 m to calculate the cross-correlation, and then adjust this parameter based on CC results. First, the fiber system noise and modulation noise in DAS data were filtered out by using a Fourier transform. According to the frequency of fiber system noise and modulation noise, a bandpass filtering at 0.5 < f < 50 Hz was used before resampling. Second, the frequency–time normalization method and spectral whitening were used to normalize the microtremor data. In addition, for consistency with nodal seismometers, we resampled the microtremor data recorded by DAS to 2.5 ms. Figure 5 displays the multi-offset CC result based on 5 h DAS and three-component microtremor data; each CC computed over a large number of 50 s long time windows of microtremor data. We compared the two types of data and found that in the CC shown in Figure 5a, the SNR of the direct wave (the events indicated by the red arrow) is higher near the virtual source point, and low in the area far away from the virtual source, which is similar to the CC based on the X- and Y-components’ nodal seismometer. This is mainly because the DAS signal is mainly sensitive to the elongation and compression along the direction of fibers; the directional sensitivity will affect the waveform of the direct wave recorded by DAS. When the position of the noise source and the orientation of cable are fixed, the farther away from the noise source, the difference between the direct wave recorded by the DAS becomes larger, and the SNR of the direct waves in CC becomes smaller. Secondly, we can see that the surface wave (the events indicated by the purple arrow) in the CC, based on DAS data, is similar to that in CC based on three-component nodal seismometer data, and the continuity of the events and the SNR ratio in CC based Z-component seismic records are higher than those calculated by X-component and Y-component seismic records. Therefore, we use the inversion results based on Z-component seismic data and DAS data for comparison in subsequent experiments. 

We take two sets of parameters to calculate the frequency phase velocity spectra: the length of receiving array and each time segment are 350 m, 20 s (Figure 6a,b), and 150 m, 5 s (Figure 6c,d). We can find that the frequency phase velocity spectra obtained from the two types of data are highly similar. Affected by low-frequency modulated noise, when we use the same parameters to calculate the frequency–phase velocity spectra, the low-frequency information obtained from DAS data is less (~2 Hz), and the surface wave dispersion curve extracted from DAS data is more likely to lack the information of deep formation. However, the SNR of high-frequency information extracted from DAS data is strong, which is probably because the spacing between DAS sensors is smaller, and the spatial resolution is higher. A further feature that needs to be noted is that as the length of the receiving array and time segment increase, DAS and nodal seismometers record more low-frequency signals, which means that long receiving array and long-time data are conducive to obtaining more low-frequency signals, which is conducive to deep formation imaging.

On this basis, MASW and the damped least-squares method were used to extract the fundamental-mode Rayleigh wave phase-velocity dispersion curves and 1-D S-wave velocity profile, respectively [47]. We used the fundamental model of the dispersion curve with the frequency of 1.8 Hz to 8 Hz to inverse the shear wave velocity of deep formation. According to the imaging results, we selected the area that the active fault may pass through and calculated the CC for 5 s with a receiving array length of 150 m using the same method. Based on this, we used the fundamental model of the dispersion curve with the frequency of 2 Hz to 15 Hz to inverse the shear wave velocity of the shallow active fault.

Figure 7 shows the S-wave inversion results within 170 m underground, obtained using DAS and nodal seismometer microtremor data. The top horizontal axis is the distance in meters, and the vertical axis is the depth in meters. The white solid line is the shear wave velocity contour within the range of ~60 m. We can see that although the resolution of the S-wave velocity model obtained from DAS data and Z-component seismic data is lower than that of the reflection seismic profile obtained from active source seismic exploration, the depth and shape of the interface obtained by the two exploration techniques are consistent, especially the interface at the depth of 60 m and 140 m, and the interface at the depth of 100 m underground on the south side. This also shows that the calculation parameters we selected are effective. On this basis, we compare the S-wave velocity models based on DAS data and Z-component seismic data. First, both Figure 7a,d have a pronounced interface at a depth of approximately 120 m. The depth of this reflector becomes shallower from north to south and changes the most at station 620 m, which is similar to the characteristics of the H/V spectral ratio curve. According to the existing reflection seismic profile XL and the S-wave velocity obtained by inversion, we interpret this interface as the bottom boundary of the Quaternary sediments. In addition, the S-wave velocity structures retrieved from the two datasets are similar. In terms of velocity variation, both inversion results can be roughly divided into three parts: 0 to 40 m underground, 40 m to 120 m underground and 120 m to 180 m underground. Taking station 540 m as the boundary, we can find that the velocity of the layer within almost 40 m underground is low and that the thickness change is small, but the variations in the S-wave velocity profile for distinct depths are different. The velocity changes faster with depth in the north and relatively slower in the south. According to the velocity contour, Figure 7a,d also show that the formation on the south side, near the station 550 m, has been uplifted. Within the range of 40 m to 120 m underground, the variation in velocity in the south is significantly greater than that in the north, and this velocity change starts from approximately station 500 m and reaches the maximum at approximately station 600 m. Within the range of 120 m to 180 m underground, the variation in thickness and S-wave velocity in the north at station 660 m is small, and that in the south is large. 

Based on this, we have carried out precise processing on the shallow area (red box area) within the range of station 424 m to 664 m (Figure 7b,e). The result also indicates that the two data processing results are highly similar, which shows that the depth of the reflector becomes shallower from north to south. Secondly, within the range of station 524 m to 564 m, the shear wave velocity inversed by DAS and nodal seismometers has changed significantly, with the strata on the south side rising and the strata on the north side falling, which is consistent with the characteristics of the normal fault inferred before. At the same time, DAS has a higher sensitivity and dynamic range; it is more vulnerable to external noise. Although the SNR of CC shown in Figure 5a is lower than the CC shown in Figure 5d, the spacing of DAS sensors is smaller than that of node seismometers and has higher lateral resolution. In an actual microtremor survey, the impact of low SNR can be reduced by densifying the optical fiber sensors. We drew the corresponding fault cartoons (Figure 7c,d) according to the difference of S-wave velocity and estimated the location of near-surface active fault FP4-3 (shown by the red solid line in Figure 7). It can be seen that DAS data with a high spatial sampling rate can obtain inversion results with a higher resolution relative to nodal seismometers, which is conducive to the identification and division of active faults.

## 5. Discussion

### 5.1. Analysis of CC Characteristics

According to the location of the shallow active fault obtained from the detection results shown in Figure 7, we reanalyzed the CC (Figure 8) and found that there are obvious events with a specified period of time (covered by red transparent strips) near the shallow active fault (approximately 540 m). The events on both sides of faults are obviously staggered, and there are also obvious differences in the apparent velocity and energy. When the virtual source is located at the footwall, the apparent velocity of the events on the north side of the diffraction point is greater than that on the south side, and vice versa. At the same time, since the data recorded by DAS represents the average axial strain over the gauge length, those signals are not an obvious hyperbola, but rather a straight line with smaller curvature.

We conducted a microtremor survey numerical simulation experiment based on two-dimensional viscous acoustic equations to study this characteristic. According to the microtremor survey results, we constructed the two-dimensional normal fault model shown in Figure 9. The model parameters are shown in Table 2. The microtremor survey numerical simulation uses 1200 random sources, whose dominant frequencies are randomly distributed from 1 Hz to 40 Hz (as shown in Figure 9b). We use the signal which was formed by the convolution of random time series and the Ricker wavelet with different frequencies and energies as the microtremor source. The numerical simulation parameters are shown in Table 3.

Figure 9c–f displays the snapshot at different times. We can see that, as time goes by, the seismic wave fields generated by seismic sources at different locations underground converge in fault; those seismic waves will generate internal multiples in the active fault (shown by the yellow circle in Figure 9c). According to Huygens principle, we can regard the multiples formed in the active fault as a new source array. The seismic wave generated by this source array will propagate along the fault to the surface and generate a diffracted wave at the fault breakpoint (shown by the red circle in Figure 9d–f). The shallower the breakpoint is, the stronger the diffracted wave energy will be. At the same time, due to the existence of different types of microtremor, these signals recorded by instruments arranged on the surface have a specified period.

Figure 10 shows the microseismic numerical simulation data and CC of shot points located at the hanging wall and footwall of the fault. We found that this kind of diffracted wave also appeared in the CC; the locations of these diffraction points are consistent with wave field analysis results, which are located near the fault breakpoint. At the same time, these signals have a specified period, and the direct wave in CC has obviously staggered here. These characteristics are consistent with CC based on actual DAS and three-component microtremor data. Figure 10c shows the numerical simulation shot gathered with different receiver spacing. We can find that the smaller the receiver spacing is, the easier the fault diffracted wave is to be identified (shown by the red circle in Figure 10c). Fortunately, the existing DAS equipment can easily realize fast data acquisition with super-dense spacing in a large range. In practical application, we can reduce the impact of low SNR by improving the space interval and coverage time of DAS sensors and combining the microtremor survey and diffraction wave imaging algorithms to achieve accurate positioning of shallow active faults. 

### 5.2. Variation of Ground Temperature and Strain near Near-Surface Active Faults

The existing research results show that the loose sediments, vegetation and other overburdens above the buried faults have high humidity, high porosity and high water content, which makes the thermal inertia of those overburdens greater than that of the bedrock. The high thermal inertia means that the fault zone will form a low temperature area compared with the bedrock, which is shown as a negative anomaly zone on the thermal infrared image [48]. At the same time, the measurements across the fault also indicate that the ground strain and temperature of the area across active faults may change; the infrared remote sensing satellite is often used to measure the surface temperature, and the location of buried active faults in a large range can be inferred by analyzing the fluctuation of surface temperature and field geological survey [49,50]. The spatial sampling rates of BOTDR and DTS are much higher than those of satellite thermal infrared data and point temperature and strain sensors, and the ground temperature and strain were measured at centimeter-level resolution, so we also use BODTR and DTS to monitor the ground strain and temperature information in this experiment. Due to the limitation of power supply equipment, BODTR and DTS record data at different times on the same day (as shown in Table 1).

Figure 11 shows the ground temperature and strain data at different locations along the optical fiber recorded by DTS and BOTDR. We can see that there are obvious low value anomalies in Figure 11a,c near station 540 m, and this low value anomaly is more obvious in the strain profile shown in Figure 11c. According to the temperature and strain data, we have calculated the maximum, minimum, mean and the difference between the maximum and minimum values (Figure 11b,d). The statistical result also indicates that the maximum temperature/strain, minimum temperature/strain, mean temperature/strain and the difference between the maximum and minimum values near the 540 m station have changed significantly. With the station 540 m as the boundary, there was clearly a minor fluctuation in temperature and strain in the south and a drastic fluctuation in the north. The consistent locus of the changes in ground temperature and strain match the inferred surface location of fault F4 well. According to the active seismic survey and microtremor survey detection results, we speculate that the main reason for this temperature difference is the thermal inertia difference caused by the thickness difference of the overburden between the hanging wall and the footwall of the active fault. The strain difference may also come from the significant differential settlements between the hanging wall and the footwall of the active fault. Meanwhile, the changes in strain are more pronounced than temperature, especially near active fault FP4-3. This may be because the temperature recorded during the experiment is the comprehensive embodiment of the land surface, air and other external factors: the temperature data are recorded for a long time (6 h); the external temperature changes greatly during the monitoring period; and it is difficult to identify the data fluctuations caused by a single change in ground temperature. The strain recorded by BOTDR is also a comprehensive reflection of the underground medium change and external factors such as meteorological factors, environmental loads and human factors. However, the strain data in this experiment are relatively short (2 h); these external conditions change relatively little in a short time, and the strain change caused by the ground is relatively obvious. Of course, this is only a conjecture about this phenomenon. There may be some undiscovered relationship between the location of shallow active faults and ground temperature and strain, and we will study this further in the future. However, this phenomenon also shows that we can simultaneously record ground temperature and strain data, so as to provide a low-cost and high spatial resolution technology for urban safety and environmental assessment.

## 6. Conclusions

In this paper, we investigated the applicability of imaging near-surface active fault zones with distributed acoustic sensing. The use of high spatial resolution DAS data to improve the imaging accuracy of microtremor surveys and long-term monitoring of ground temperature and strain measurements to constrain the underground detection results is expected. 

The experimental results suggest that in the near-surface active faults exploration, the microtremor survey results-based DAS data are similar to those obtained from conventional nodal seismometers. Affected by the directional sensitivity and DAS system noise, the SNR of the low-frequency surface wave signal received by DAS is weaker than that of the nodal seismometers. However, because the DAS signal is mainly sensitive to the elongation and compression along the direction of fibers, which is conducive to the reception of high-frequency surface waves, especially high-order surface waves, this allows us to obtain shallow active fault information with higher resolution. At the same time, the research results of this paper show that dense geometries are conducive to improving the imaging accuracy of microtremor seismic surveys; DAS can realize high density spatial sampling at a low cost, which is conducive to the exploration of near-surface active faults in a large range. How this advantage can be used to improve the accuracy of shallow active fault tomography is worthy of further exploration in the future.

Secondly, comparing the results of active fault exploration by DAS and the long-term monitoring results of ground temperature and strain by BOTDR and DTS, it is not difficult to find that the location of the largest change in ground temperature and strain is in agreement with the location of the near-surface active faults obtained from the inversion results of microtremor sources. This shows that in practical exploration, we can comprehensively use the high-spatial resolution ground temperature and strain data recorded by BOTDR and DTS, as well as the underground formation information detected by DAS to estimate the location of near-surface active faults, so as to reduce the influence of human interference and lack of prior information on the detection results, and improve the reliability of the detection results.

Another feature to note is the CC profile. Abukrat et al. [32] showed that the void on the shallow surface would form a backscattered signal, and the shallow diffraction imaging was applied to identify the location of the void. Combined with the exploration results and numerical simulation results, we re-analyze the multi-offset CC calculated with DAS and nodal seismometers microtremor data, and found that there were obvious scattered waves that appeared on the CC profile near the active fault. In future research, we may try to extract the scattered wave for active fault imaging.

## Figures and Tables

**Figure 1 ijerph-20-02915-f001:**
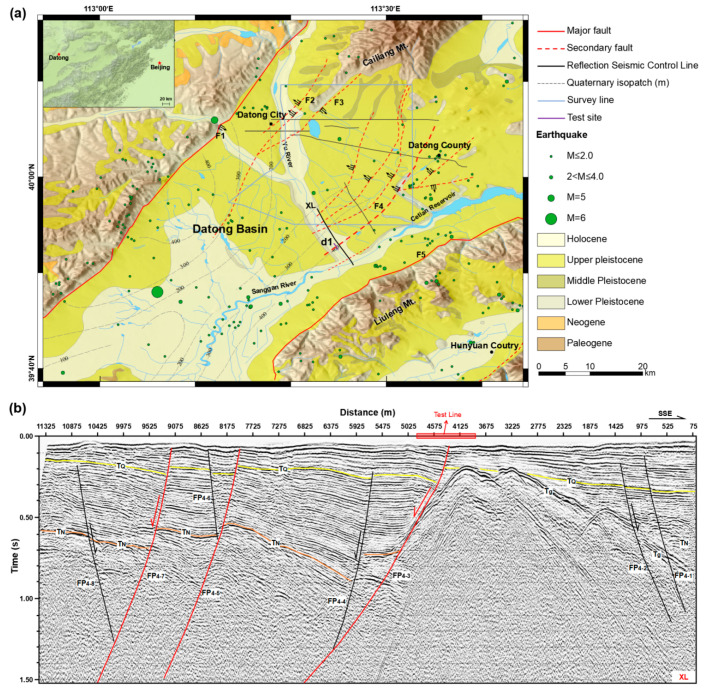
Geological maps of Datong basin. The active faults detected in this paper are marked as F4 with a bold red dashed line. The distributed optical fiber sensing at the Yu River site is denoted as the purple bar called d1 in (**a**) near line XL crossing fault F4, (**b**) shows the active seismic survey detection result of line XL, approximately 300 m to the south of line d1.

**Figure 2 ijerph-20-02915-f002:**
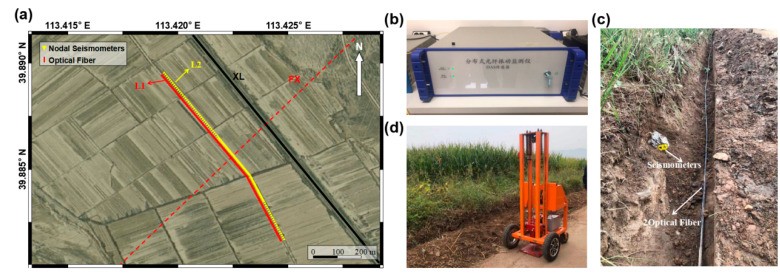
Scene of field deployments of the distributed optical fiber and seismometers. (**a**) shows the location of optical fiber and nodal seismometers line; the ground of the test site is flat with little fluctuation. (**b**) shows the DAS interrogator used in this paper. (**c**) is the horizontal trenching of a DAS cable and three-component nodal seismometers at the surface. (**d**) is the small tamping machine used in this experiment.

**Figure 3 ijerph-20-02915-f003:**
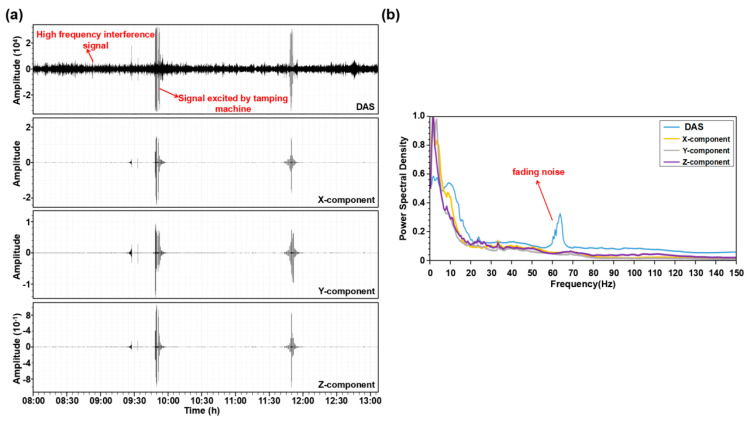
Raw data recorded by DAS and nodal seismometer. (**a**) Piece (5 h) of raw DAS and three-component nodal seismometer microtremor data. (**b**) displays average noise spectra computed over a large number of 2 s long time windows of microtremor data.

**Figure 4 ijerph-20-02915-f004:**
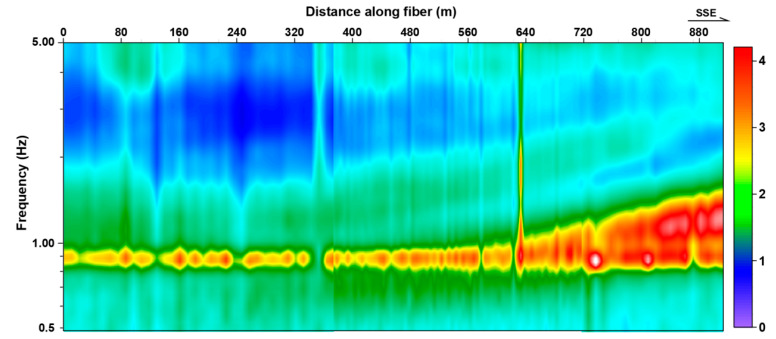
H/V spectral ratio curves calculated by Geopsy based on the three-component seismometer data.

**Figure 5 ijerph-20-02915-f005:**
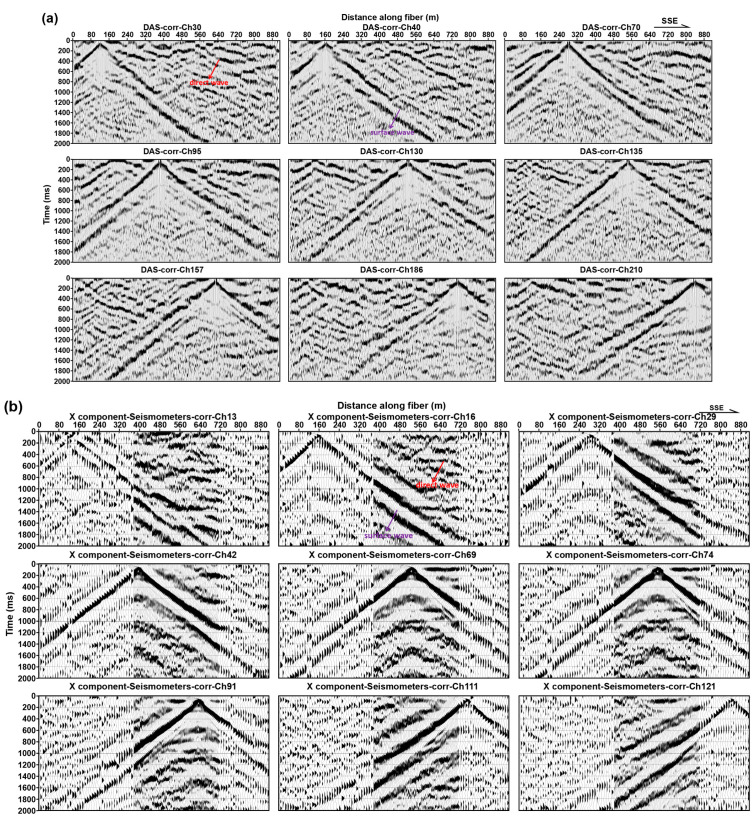
Examples of the multi-offset CC result based on 5 h DAS (**a**) and three-component microtremor data (**b**–**d**), respectively. To increase the SNR ratio, we stack the casual and acasual traces linearly.

**Figure 6 ijerph-20-02915-f006:**
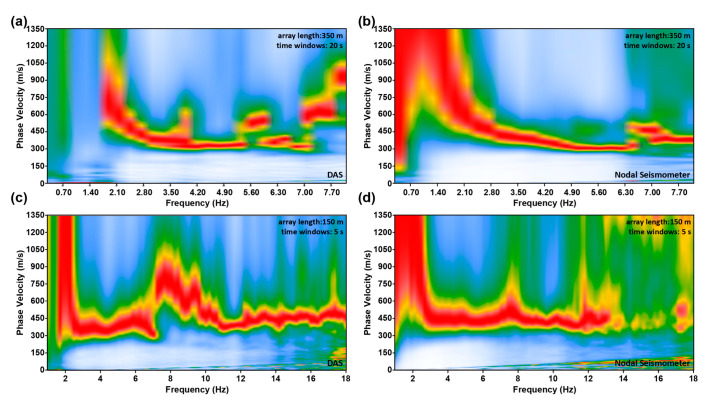
Examples of the frequency–phase velocity spectra based on DAS (**a**,**c**) and nodal seismometer (**b**,**d**) data. The virtual source point is located near 352 m, the length of receiving array and each time segment are 350 m, 20 s (**a**,**b**), and 150 m, 5 s (**c**,**d**).

**Figure 7 ijerph-20-02915-f007:**
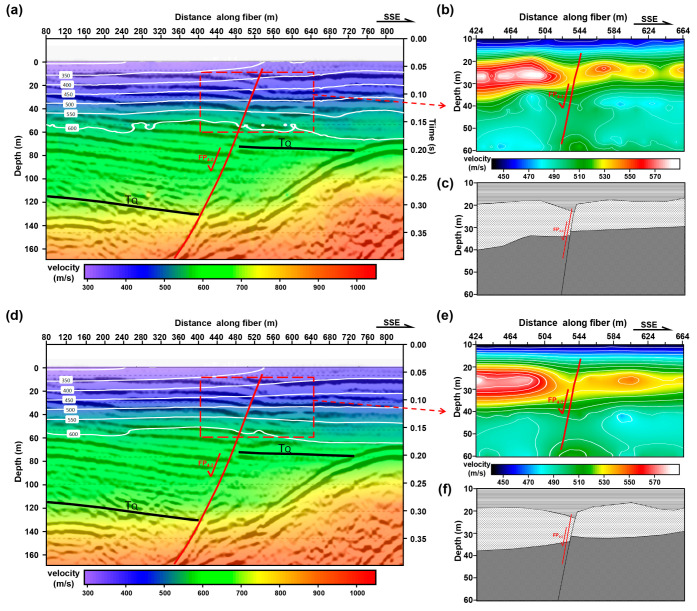
Microtremor Survey detection results. (**a**) and (**d**) are S-wave inversion profiles based on DAS data; these results use the dispersion curve information with the frequency of 1.8 Hz to 8 Hz and 2 Hz to 15 Hz, respectively. The wiggles in (**a**) and (**d**) are the active seismic survey detection results of line XL, shown in Figure 1b. (**c**) is a simple fault cartoon drawn according to the inversion results shown in (**b**). (**d**) and (**e**) are S-wave inversion profiles based on Z-component seismic data using the same method and parameters. (**f**) is a simple fault cartoon drawn according to the inversion results shown in (**e**); the color bars represent the velocity value of S-wave.

**Figure 8 ijerph-20-02915-f008:**
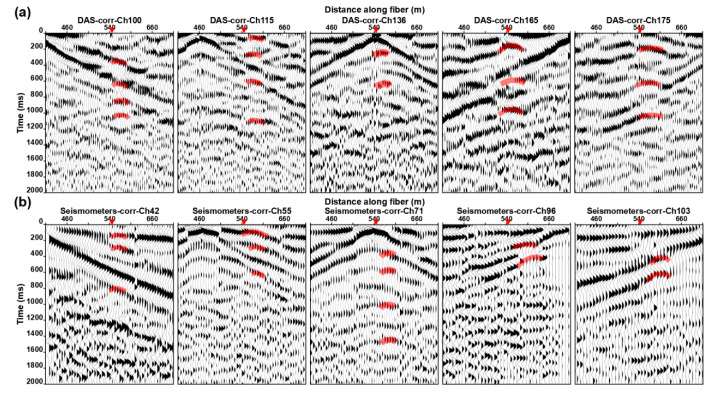
Examples of the multi-offset CC result based on 5 h DAS (**a**) and nodal seismometers data (**b**), respectively. The virtual source point is located near active fault.

**Figure 9 ijerph-20-02915-f009:**
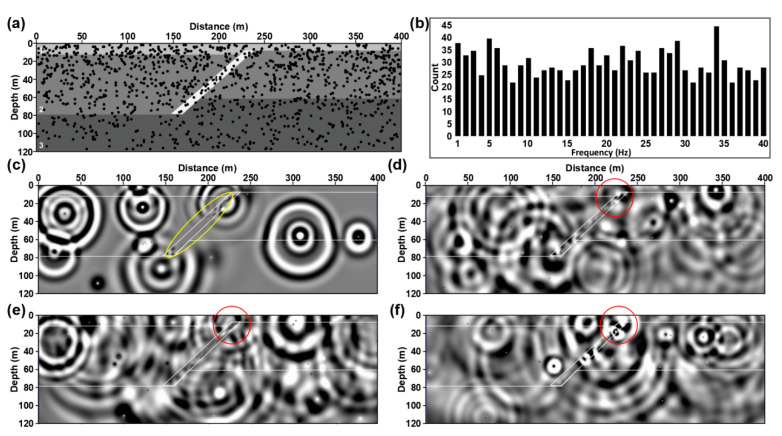
Display of numerical simulation. (**a**) is a map view of the two-dimensional normal fault model. (**b**) is the histogram of dominant frequency distribution of source. (**c**–**f**) are wave field snapshots at different times.

**Figure 10 ijerph-20-02915-f010:**
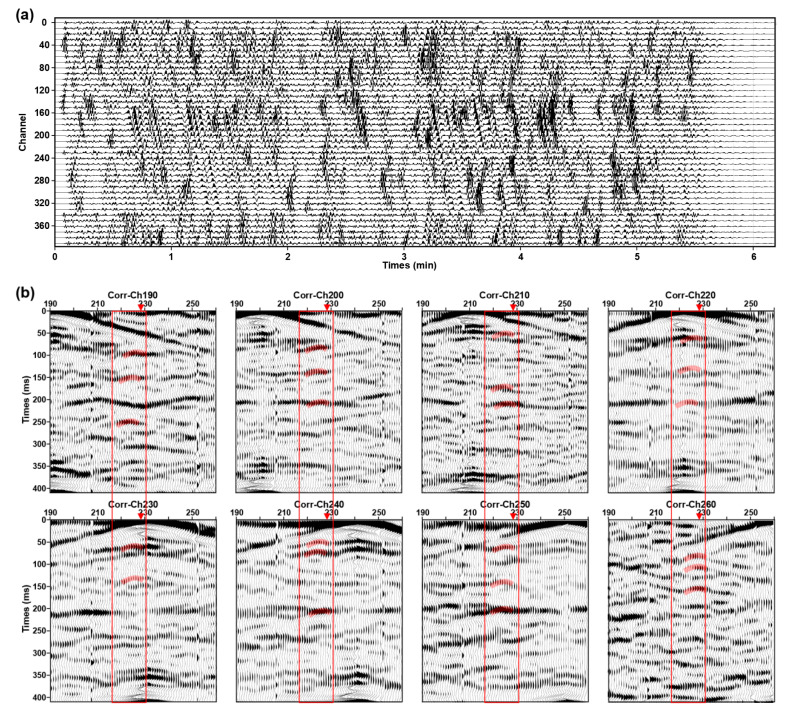
Display of microtremor numerical simulation results. (**a**) is numerical simulation microtremor data. (**b**) are CC with 1 m receiving point spacing. The virtual sources are located at the hanging wall and footwall of the normal fault. (**c**) are CC with different receiver spacing.

**Figure 11 ijerph-20-02915-f011:**
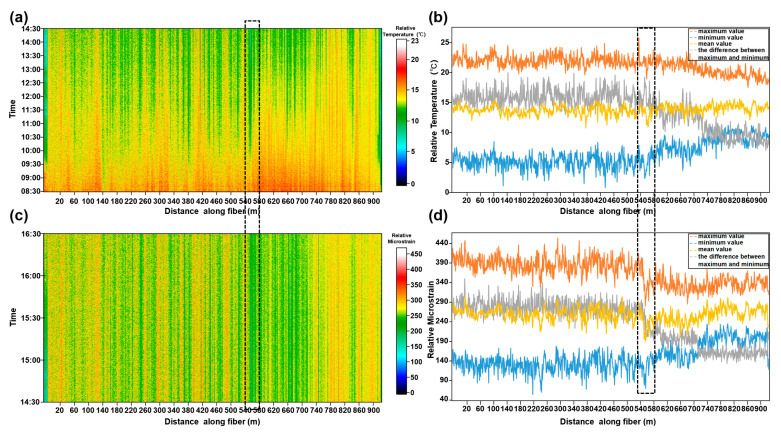
Monitoring results of BOTDR and DTS. (**a**) and (**c**) show the variation in the relative temperature and microstrain of the land surface; the color bars are the relative temperature and the relative microstrain value, respectively. (**b**) and (**d**) are the maximum temperature/microstrain, minimum temperature/strain, mean temperature/strain and the difference between the maximum and minimum values at different positions of the optical fiber during the measurement time. The location shown in the black dashed box is the area with the largest value change, and also the location of shallow active fault.

**Table 1 ijerph-20-02915-t001:** Geometric parameters of the microtremor survey.

Instrument	Station Number	Interval (m)	Total Channel	Sampling (ms)	Record Time (h)
**Three-Component Seismometers**	**1–20**	**10**	**128**	**2.5**	**6**
	**21–90**	**5**			
**DAS/BOTDR/DTS**	**90–128**	**10**	**250**	**0.125/5000/5000**	**5.1/2/6**
	**1–250**	**4**			

**Table 2 ijerph-20-02915-t002:** Parameters of two-dimensional normal fault model.

Layer	V (m·s^−1^)	Rou (kg·m^−3^)	Q
1	700	1.5	40
2	832	1.7	50
3	1000	1.82	55
fault	340	1.2	35

**Table 3 ijerph-20-02915-t003:** Numerical parameters of the microtremor survey numerical simulation.

Number of Sources	Dominant Frequency of Source (Hz)	Sampling (ms)	Record Time (min)
1200	Min: 1 Max:40	2	6

## Data Availability

The data that support the findings of this study are available from the corresponding author, Junjie Ren, upon reasonable request.

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
