# Peer review of "Distributed Acoustic Sensing Based on Microtremor Survey Method for Near-Surface Active Faults Exploration: A Case Study in Datong Basin, China"

_ijerph, 2023, doi:10.3390/ijerph20042915_

Round 1
Reviewer 1 Report
1. In the Data Analysis and Processing, the fiber system noise and modulation noise in DAS data were filtered out using a Fourier transform. What is cut-off frequency in the used filter?
2. In Figure 5, authors compared the multi-offset cross-correlation results based on the DAS and the vertical component nodal seismometer. Since DAS is sensitive to the elongation and compression along the direction of fibers, why did the author choose the vertical component of geophone to compare with DAS? What are the cross-correlation results based on the X and Y components nodal seismometer?
3. H/V spectral ratio curves are calculated using the three-component seismometer data. According to the H/V spectral ratio curves shown in Figure 4, authors processed the DAS data. How is the prior information of H/V spectral ratio curves used for the subsequent DAS data processing?
4. In the Introduction, authors summarized that DAS has obtained good results in the seismology community such as in earthquake detection, hydrocarbon and geothermal exploration, urban monitoring, volcano and glacier monitoring, subsurface imaging and other seismic activities. In addition to being used in the above aspects, DAS is also applied to the real-time monitoring of microseismic and fluid injection in hydraulic fracturing (Zheng et al., 2022). For the application of DAS in subsurface imaging, it is better to add the recent researches of DAS in near-surface imaging (Shao et al., 2022a, b).
Zheng, et al. 2022. A deep learning approach for signal identification in the fluid injection process during hydraulic fracturing using distributed acoustic sensing data. Frontiers in Earth Science, 1820.
Shao, et al. 2022a. Near-surface characterization using high-speed train seismic data recorded by a distributed acoustic sensing array. IEEE Transactions on Geoscience and Remote Sensing, 60, 1-11.
Shao, et al. 2022b. Near-surface characterization using urban traffic noise recorded by fiber-optic distributed acoustic sensing. Fronties in Earth Science, 10, 943424.
5. In addition to the DAS and nodal seismometers, DTS is also used to estimate the location of near-surface active faults. Why is the change of temperature related to faults?
Reviewer 2 Report
This manuscript provides a real data experiment of using DAS data to study and monitor the active fault via cross-correlation of surface waves. The authors make a thorough comparison of seismometers with DAS recording in terms of the data quality, CC functions and dispersion curve inversion results. I would recommend acceptance after major revision. Here are my comments
1. How do you make sure that the signals marked by red in Fig 8 and Fig.10(b) are from scattering? Did you use any criterion or simply based on your experience of dealing with field data. Please describe this in the manuscript.
2. What are the wiggles in Fig7a,b? How are these wiggles related to your velocity model, Please elaborate in the manuscript.
3. The authors analyze the velocity model in Line 190-210, but it is not intuitive enough for readers to understand based on these dry words. I would suggest you draw a simple cartoon, showing your inferred fault location plus three layers of sedimentation with dipping, so that readers will easily understand the geological setting here.
4. Add reference to the :multichannel surface wave analysis” in line 67
5. Fig. 11. What physical quantities are you showing for the rightmost colorbars? Please add it nearby those colorbars, and explain it in the caption.
6. For section 5.2, I feel like these analysis is not quantitative enough. I can get your point of “smaller fluctuation at South, larger fluctuation at North”, but your discussion is not deep and quantitative. There is even no number related to the temperature and strain shown in Fig. 11
7. The authors need to rephrase line 220-223 to make it more concise. And also, delete one “therefore” in line 291
Round 2
Reviewer 2 Report
Dear Authors,
Thanks for bring this revision. I see significant improvement in terms of figure quality, data presentation and analysis. And I believe this will benefit readers a lot to understand the full story and conclusion of this manuscript. Your response to reviewers is well wriiten, clearly answered all the questions we have. I believe this version is ready for publication.